# A Review on the Role of Food-Derived Bioactive Molecules and the Microbiota–Gut–Brain Axis in Satiety Regulation

**DOI:** 10.3390/nu13020632

**Published:** 2021-02-16

**Authors:** Nuria A. Pizarroso, Pablo Fuciños, Catarina Gonçalves, Lorenzo Pastrana, Isabel R. Amado

**Affiliations:** International Iberian Nanotechnology Laboratory, Av. Mestre José Veiga s/ n, 4715-330 Braga, Portugal; nuriaalpi07@gmail.com (N.A.P.); pablo.fucinos@inl.int (P.F.); catarina.goncalves@inl.int (C.G.); lorenzo.pastrana@inl.int (L.P.)

**Keywords:** gut–brain axis, bioactive peptides, obesity, probiotics, in vitro models, prebiotics, intestine, hormones

## Abstract

Obesity is a chronic disease resulting from an imbalance between energy intake and expenditure. The growing relevance of this metabolic disease lies in its association with other comorbidities. Obesity is a multifaceted disease where intestinal hormones such as cholecystokinin (CCK), glucagon-like peptide 1 (GLP-1), and peptide YY (PYY), produced by enteroendocrine cells (EECs), have a pivotal role as signaling systems. Receptors for these hormones have been identified in the gut and different brain regions, highlighting the interconnection between gut and brain in satiation mechanisms. The intestinal microbiota (IM), directly interacting with EECs, can be modulated by the diet by providing specific nutrients that induce environmental changes in the gut ecosystem. Therefore, macronutrients may trigger the microbiota–gut–brain axis (MGBA) through mechanisms including specific nutrient-sensing receptors in EECs, inducing the secretion of specific hormones that lead to decreased appetite or increased energy expenditure. Designing drugs/functional foods based in bioactive compounds exploiting these nutrient-sensing mechanisms may offer an alternative treatment for obesity and/or associated metabolic diseases. Organ-on-a-chip technology represents a suitable approach to model multi-organ communication that can provide a robust platform for studying the potential of these compounds as modulators of the MGBA.

## 1. Introduction

Physiologically, there is an energetic balance between caloric intake, controlled by satiety and hunger, and caloric expenditure. The current lifestyle promotes sedentary and abusive consumption of fast food and alcohol, leading to energy accumulation, as well as to increased risk of developing obesity and overweight [1]. According to the WHO, both diseases affected almost 3 billion people in 2016 [2]. Obesity is intimately related to the development of non-communicable chronic diseases (NCDs), such as type 2 diabetes (T2DM) and insulin resistance, dyslipidaemia, hypertension, and cancer. Common comorbidities also include hepatic, neurodegenerative and inflammatory diseases, and depression [3,4]. As a consequence of the development of these concomitant diseases, obesity-related morbidity and mortality have increased significantly in the last years.

The impact of an imbalanced diet includes a series of physiological changes in obese and overweight individuals, including modifications in the profile of gut hormone secretion that produce alterations in the endogenous satiating pathways. Moreover, the loss of beneficial microorganisms replaced by pathogenic intestinal flora, known as dysbiosis, is caused mainly by imbalanced diets [4,5,6] and the administration of antibiotics [7]. The role of the microbiota–gut–brain axis (MGBA) in the regulation of appetite has become a central topic in recent years since gut hormones are known as mediators of satiation signals. The satiating mechanisms involve orexigenic hormones (appetite stimulators) and anorexigenic hormones (appetite inhibitors) that regulate food intake, stimulating the release of other hormones and neuropeptides [8]. These appetite inhibitors or satiety stimulators have attracted considerable interest in addressing new treatments for obesity and overweight. In particular, food-derived compounds are regarded in the literature as interesting molecules to take advantage of their own natural and innate effects on food intake regulation.

This review aims to gather relevant existing knowledge on the MGBA regulation, including intestinal hormone release and receptors involved, and the pathways implicated in generating satiety feelings in the brain including the role of microbiota. Moreover, to pull together a critical review of food-derived bioactive molecules showing in vitro and/or in vivo endocrine modulatory activity, along with the future challenges of in vitro models to study the MGBA regulation.

## 2. Gut–Brain Axis: Endogenous Mechanism of Food Intake Regulation 

The gastrointestinal tract (GIT) and the brain interact in a complex and bidirectional way, using blood circulation and cranial nerves as communication channels. In this process, hormones released by the enteroendocrine cells (EECs) stimulate the brain through the humoral or the neural pathway (Figure 1). 

The EECs act as sensors of the luminal content, releasing different combinations of intestinal hormones, which act synergistically to stop food intake and produce satiety through endocrine and nervous signaling (Figure 2). The intestinal hormones, including cholecystokinin (CCK), glucagon-like peptide 1 (GLP-1), peptide YY (PYY), oxyntomodulin (OXM), and gluten immunogenic peptides (GIP) arrive at the brain (afferent pathway) and generate central signals (afferent pathway) that transmit impulses to the peripheral organs causing an effect (efferent pathway). CCK is involved in different physiological processes, such as intestinal motility, gastric emptying, gallbladder contraction, and pancreatic secretion, but also food intake regulation and energy homeostasis. CCK negatively regulates the secretion of Ghrelin, the unique orexigenic hormone. GLP-1, the product of pre-proglucagon gene cleavage, has anorexigenic and hypoglycaemic activity [8,9], inhibits glucagon secretion, and stimulates insulin synthesis and release. This incretin also inhibits food-induced gastric acid secretion, promotes gastric distention, and delays gastric emptying, generating satiety feelings. GLP-1 has a half-life of around two minutes in the plasma because the plasmatic enzyme dipeptidyl peptidase IV (DPP-IV) hydrolyses the peptide [10]. PYY promotes the absorption of nutrients by delaying gastric emptying (“ileal brake”) and inhibits fluid and electrolyte secretion in the small intestine. PYY also has vasoconstrictor activity and participates in postprandial natriuresis, reducing the speed of glomerular filtration [11]. 

In the humoral pathway, intestinal hormones released into the bloodstream have an endocrine effect on the hypothalamus and brainstem, both involved in satiation mechanisms. The access to both brain regions occurs by specific receptors CCK-2R, GLP-1R, and Y2 through the highly vascularized areas—median eminence and postrema. The arcuate nucleus (ARC), as the most significant hypothalamic nucleus in satiety response, contains two types of neurons. One of the neuron populations contains neuropeptides (central signals) such as pro-opiomelanocortin (POMC) and cocaine- and amphetamine-regulated transcript (CART), having agonist effects on melanocortin 3 (MC3) and 4 (MC4), inducing satiety. The other population is rich in neuropeptide Y (NPY) and agouti-related peptide (AgRP), which produce food intake stimulation. In the brainstem, there are also neuronal populations such as POMC neurons at the nucleus tractus solitarius (NTS). In the neural pathway, hormonal release has a paracrine effect on the vagal nerves that connect the GIT bidirectionally to the NTS in the dorsal vagal complex (DVC) of the brainstem, where, as summarized above, satiety feelings appear. Recently, the discovery of the complete morphology of EECs revealed a more intimate and complex interaction between EECs and vagal afferent neurons, as suggested initially through an axon-like process, known as neuropods [11,12,13]. The presence of intestinal hormone receptors in the vagal afferent nerves, such as CCK-1R, GLP-1R, and Y2 receptor, was also reported [11,12,13]. The recent report of the transport of specific metabolites from the luminal content through the EECs to the lamina propria must also be highlighted, wherein they can directly stimulate the nerve endings of the vagal afferent nerves [11].

The luminal digestive content is the major stimuli of intestinal hormone secretion although the predominant mechanism by which it generates satiety signals in the brain is under debate—the short half-life of some hormones, such as GLP-1, challenges the humoral pathway as a primary mechanism. Nevertheless, only 30–50% of EECs appear to have neuropods (neural pathway), which suggests a combination of both mechanisms. Namely, there is a rapid vagal response due to neurotransmitters release, while intestinal hormonal stimulation contributes to a lengthy response [11,14].

EECs are histologically classified in different subtypes, being I and L cells responsible for CCK release, and GLP-1 and PYY release, respectively. The G-protein-coupled receptors (GPCRs) are activated by macronutrients, acting as secretagogues of peripherical signals but also as transcriptional activators of gut hormone genes [15]. The GPCR expression occurs throughout the GIT in the apical and basolateral cellular membrane, the latter occurring by paracellular transport [16,17]. Different GPCRs respond to specific food-derived metabolites, as summarized in Table 1. Most receptors sensitive to food luminal contents act through coupling to intracellular G-proteins, such as Gαi, Gαq, and Gαs, through activation of second messengers [18]. The intracellular signaling pathways involved in the transcriptional activation of intestinal hormone genes and their release from different receptors is still unclear. What it is known is that hormone secretion occurs from the basolateral side of the EECs to the lamina propria, where they have a paracrine and/or endocrine effect [13]. 

The intracellular signaling cascade implicates both secondary messengers, such as cAMP, Ca^2+^, or IP_3_, and intracellular proteins, such as MEK, PKA, or CaMK [15,34]. Nonetheless, intracellular Ca^2+^ mobilization has a critical role not only in the secretion of satiating hormones to the lamina propria but also in the transcriptional activation of hormone genes [34,35]. 

The dysregulation of intestinal hormone secretion arises not only in obese and overweight patients but also in people suffering from T2DM. For instance, plasmatic GLP-1 and PYY are lower in those people, due to their reduced secretion in the gut [36]. Recently, DPP-IV was identified as an adipokine with high production in the adipose tissue of obese people, inducing increased hydrolysis of GLP-1 and OXM, reducing their half-life and producing a lower satiating effect [37]. The CCK plasmatic level is higher in obese people, even though these patients do not experience more evident satiety feelings, due to receptor desensitization present in brain areas involved in satiation [38]. These physiological modifications affecting the mechanisms of satiation in patients who suffer from obesity and overweight are challenging to understand, hindering the search for new treatment targets. 

## 3. Intestinal Microbiota: A Significant Player in Satiety

The intestinal microbiota (IM) provides the host with numerous benefits, such as an effective immune system and essential vitamins, and is also critical for the digestion of some fats, proteins, carbohydrates, and fibers of the diet. Bacterial metabolites can also have protective effects against various diseases, such as colorectal cancer, inflammatory bowel diseases, and obesity. Contrarily, microbial metabolic products of pathogenic strains could contribute to the pathogenesis and/or progression of diseases [39]. 

Numerous studies report the relationship of the IM to multiple metabolic disorders [40]. In obesity, this condition links to alterations in the relative proportion of different microbial groups, an overall reduction of microbial diversity, and altered representation of genes and bacterial metabolic pathways [41,42]. Different studies reported elevated levels of *Firmicutes* and reduction of *Bacteroidetes*, increased *Archaea* counts, reduction in *Bifidobacterium,* and increased *Halomonas* and *Sphingomonas* in animal models of obesity (rat and mice) [43]. Similar alterations occur in the microbiota of obese humans, i.e. increased *Firmicutes*/*Bacteroidetes* ratio [44], a higher proportion of *Actinobacteria* [41], and reduced numbers of *Bifidobacteria* [45]. On the other hand, trials combining a restriction of caloric intake with physical activity showed an increase in *Bacteroides*/*Prevotella* levels and a decrease in *Clostridium* coupled to weight loss [46]. The role of IM in obesity is extremely multifaceted, playing an essential role in digestion and energy storage. The IM provides the host with key enzymes for different metabolic processes, including complex polysaccharide catabolism, vitamin and amino acid biosynthesis, and bile acid deconjugation and dehydroxylation, which control lipid solubilization and absorption [43]. For this reason, as a whole, it is assumed that IM increases host ability to extract energy from food and promotes fat storage [47,48,49]. The IM also participates in the regulation of the energy balance through the interaction with the central nervous system, acting on the synthesis and activity of hormones and neuropeptides [43]. 

Diet is the primary factor in the modulation of the composition of the IM by providing specific nutrient sources and inducing environmental changes in the gut ecosystem (Figure 3). A high-fat diet reduces diversity, the total number of microbes, but increases commensal and pathogenic Gram-negative bacteria, associated with obesity, such as *Proteobacteria* and *Deferribacteres* [6]. The increase of these detrimental bacteria leads to higher amounts of lipopolysaccharide (LPS), which are a triggering factor for low-grade inflammation in obesity [5,6]. Bacterial-induced inflammation is derived from LPS increase but also intestinal barrier-enhanced permeability, producing chain reactions with adverse effects [50]. LPS, along with flagellin or peptidoglycans, are translocated from the intestinal lumen to the lamina propria through the intestinal epithelium in obesity disease. LPS can desensitize the vagal afferent nerves in the lamina propria. Moreover, LPS leads to attenuated CCK-induced satiation and dysregulation of anorexigenic and orexigenic hormones expressed in vagal afferent neurons, concomitant with hyperphagia and obesity development [51]. 

The type and number of microbial metabolic products depend on the quality and quantity of the IM, also determining their effects on the host. Several bacterial metabolites have an essential impact on the host central nervous system, including satiety, suggesting they might be neuroactive compounds. For instance, short-chain fatty acids (SCFA) are bacterial metabolites produced from indigestible protein and carbohydrate fermentation. The most common SCFA are butyrate, propionate, and acetate, associated with anti-inflammatory responses in the colon, and they reduce food intake by triggering intracellular signaling to release anorexigenic peptides. Butyrate and propionate have been demonstrated to have a protective role against diet-induced obesity and insulin resistance [31]. Receptors for SCFA are found in neurons, suggesting a direct stimulation on the vagal afferent nerves [5]. 

Moreover, bacterial proteins have been linked to satiety signaling. Recent reports found that the *Escherichia coli* caseinolytic protease B (ClpB), identified as a conformational mimetic of the anorexigenic neuropeptide α-melanocyte-stimulating hormone (α-MSH), stimulates the secretion of PYY in primary cell cultures of rat intestinal mucosa at nanomolar concentrations [52]. These results were not found in ClpB-deficient *E. coli* strains suggesting a link between ClpB and satiety signaling in the gut. Further studies revealed that the commensal bacteria *Hafnia alvei*, synthetizing the ClpB protein with an α-MSH-like motif, reduced food intake, body weight gain, and fat mass in mice obesity models when administered by oral gavage for 18 and 46 days [53]. These results are a preclinical validation of the use of the strain *Hafnia alvei* as a probiotic in the management of obesity and overweight.

Diet intervention is a useful strategy to manipulate microbial diversity, composition, and stability to maintain a balanced IM, providing the host with the beneficial effects described above. Nonetheless, fiber-rich diets (prebiotics) solely cannot restore microbial diversity, but further fiber-fermenting bacteria, such as probiotics, are needed. Indeed, the administration of pre- and probiotics is the standard dysbiosis treatment to enhance the growth of beneficial bacteria. 

In summary, in pathological situations, such as obesity, beneficial bacteria and satiety-inducing metabolites decrease together with an increase of pathological bacteria and inflammatory metabolites worsen the disease. Both alterations in hormonal secretion and desensitization of afferent vagal nerves are responsible for the altered satiation mechanisms in obesity. Therefore, the implantation of beneficial microorganisms (probiotics) and their substrates (prebiotics) could be harnessed as a coadjuvant treatment of obesity, as other authors suggest, for gastrointestinal diseases, such as inflammatory bowel disease and ulcerative colitis [54,55,56]. Nevertheless, there is some debate about the actual efficacy of this strategy for modulating the gut microbiota in obese/overweight individuals. Indeed, a recent systematic review of 24 randomized controlled trials (1587 participants) concludes probiotics have minimal effects on appetite-related hormones [57]. However, the increasing number of reports supporting the functionality of pre- and probiotics in modulating IM composition might contribute to stablishing the benefits of probiotics for this purpose in obese and overweight people. A more detailed description of the role of IM in obesity and some examples of dietary interventions with pre- and probiotics are given in Section 4.2. 

## 4. Food-Derived Bioactive Molecules as New Active Compounds Stimulating Hormone Secretion

Understanding the mechanisms that interconnect food digestion, microbiota, nutrient sensing at the gastrointestinal tract, and satiety signaling is essential to design novel treatments targeting obesity. In this section, we review some of the existing knowledge regarding how dietary interventions with different macronutrients (proteins/peptides, carbohydrates, and fats) affect the gastrointestinal hormone secretion involved in satiety signaling, as well as their effects in obesity control.

### 4.1. Proteins and Peptides

Proteins, both ingested in the diet and endogenous, are digested in the small intestine by pancreatic enzymes and peptidases [5]. Then, transporters present in the brush borders and basolateral membranes of the enterocytes carry a significant amount of oligopeptides and amino acids from the lumen to the portal bloodstream [58]. Once absorbed, they can serve as the building blocks for protein synthesis, but they can also act as precursors of different metabolites in reactions involving the intestinal mucosa and microbiota [59]. Amino acid transporters expressed in the intestine can be found from the duodenum to the ileum, with the highest density in the jejunum [60]. Minute absorption occurs in the colon since colonocytes do not absorb significant amounts of amino acids, and, thus, residual quantities are fermented by resident bacteria to different metabolic end-products [61]. Some intestinal bacteria can also synthesize specific amino acids, implying that there could be a bidirectional exchange of amino acids between the host and the intestinal microbiota [59]. There is, therefore, a complex communication/interaction between protein intake (diet), host, and intestinal microbiota whose mechanisms are still under study. 

In this context, the role of dietary proteins in the mechanisms involved in satiation is the focus of a large number of studies. There is evidence that diets with high protein content reduce food intake, induce satiety, and increase energy expenditure compared to lower protein-containing diets [62,63]. Satiating properties of proteins are due to several physiological effects, including the stimulation of gut hormone secretion and the indirect stimulation of gluconeogenesis through the interaction of food-derived peptides with peripheral opioid receptors [64]. However, the mechanisms leading to the release of peripheral signals mediating in the short-term regulation of food intake are still unclear. Although whole proteins are known for their satiating effects, there is also a consensus that gastrointestinal digestion is necessary for proteins to exert or enhance this activity. Protein degradation that occurs during digestion produces the release of bioactive peptides that might have significant physiological effects on energy homeostasis [64]. However, quantifying this activity is complex and makes it challenging to identify which proteins have a greater or less satiating effect. There is also debate about whether the satiating effect of proteins also depends on the protein source, and there is more and more research trying to elucidate this aspect of food protein satiation. 

Amongst the most studied food proteins with observed effects in satiety and food control are dairy products, in particular, whey. This dairy protein has generated interest in different fields of research due to it being an inexpensive source of high nutritional quality protein. Whey protein affects satiation through the action of either intact whey protein fractions, bioactive peptides derived from whey, amino acid products of its digestion, and the combined action of whey protein or its products with other milk constituents [65]. Bioactive peptides from whey released during food processing (by enzymatic hydrolysis or fermentation) or during gastrointestinal digestion have received particular interest due to their anti-hypertensive and opioid activity [66,67,68]. Casein is another abundant milk protein known for its beneficial role in the regulation of food intake, postprandial glycaemia, and enteroendocrine hormone secretion. A recent study evaluated casein and whey protein gastrointestinal digests as inducers of CCK and GLP-1 secretion in STC-1 cells [69]. The findings revealed intestinal digestion products of both proteins, containing small-sized peptides and free amino acids, to be more potent CCK inducers than digests from the gastric phase. They also found GLP-1 release was higher with casein gastric digests and whey protein intestinal digests.

Interestingly, these authors also concluded that whey and casein hydrolysates increased CCK and GLP-1 gene expression in STC-1 cell line. Several other reports have agreed that dairy proteins stimulate peripheral hormone secretion in cell cultures and in vivo [70,71,72]. Recently, increased levels of specific amino acids (isoleucine, leucine, lysine, methionine, phenylalanine, proline, tyrosine, and valine) have been positively correlated with higher GLP-1 levels and satiety, and negatively with hunger after ingestion of an isocaloric drink containing whey proteins, in a study in obese female subjects [72]. Dietary amino acids, such as tryptophan, glutamine, methionine, and branched-chain amino acids, have a function on the improvement of gut microbiota and intestinal mucosa immunity, giving rise to new dietary/clinical interventions with a positive impact in the gut–microbiome–immune axis [59]. 

Fish proteins, in particular, blue whiting hydrolysates, have shown CCK- and GLP-1-enhancing activity in STC-1 cell line [73,74]. These peptide mixtures demonstrate in vivo activity when administered to rats, increasing plasmatic levels of both hormones, reducing short-term food intake, and decreasing body weight gain [74]. A randomized clinical study reported that supplementation with fish protein hydrolysates (FPH) of a mildly hypocaloric diet improved body weight composition and increased blood concentration of CCK and GLP-1 in subjects treated with the FPH Slimpro [75]. Other hydrolysates from animal-derived proteins such as egg, pork, chicken, and beef have shown increased CCK release in STC-1 in vitro models [71,76,77]. Moreover, pork peptones produced the suppression of appetite in animal models [76]. However, there is still a lack of knowledge on the action mechanisms of these peptides in terms of their interaction with cellular and molecular targets involved in insulin and glucose metabolism [78]. Further studies need to confirm whether food supplementation with these types of food-derived ingredients could have a real impact on the obesity treatment.

Vegetable-derived peptides, and among them, soy peptides, have been studied due to promising results on their satiating effects. A β-conglycinin peptone produced by pepsin proteolysis suppressed food intake in male Sprague-Dawley rats and inhibited gastric emptying associated with CCK-increased plasma levels [79]. In this paper, the authors also concluded that β-conglycinin peptone interacts directly with the intestinal mucosal cells in the lumen to stimulate CCK release. A later study confirmed this hypothesis after identifying the fragment (residues 51 to 63 of the β-conglycinin β-subunit) with the most potent binding activity to rat intestinal brush border membrane components in vitro, and correlated these results with reduced food intake and significantly increased CCK concentration after intraduodenal infusion of this peptidic fragment in rats [80]. The mechanism underlying CCK stimulation of this soybean ß-conglycinin peptide involves calcium-sensing receptor (CaSR), which acts as a peptide sensor in enteroendocrine STC-1 cells [77,81]. When further exploring this mechanism, the authors did not find correlation between the molecular weight of ß-conglycinin hydrolysates and CaSR- or CCK-releasing activity, suggesting this receptor senses various dietary peptides independently of their molecular weight [77]. The involvement of Gαq-coupled receptor in β-conglycinin peptide sensing in enteroendocrine cells was also suggested [82]. 

Trypsin inhibitors are another group of vegetable-derived molecules studied for their anti-obesity activity. For instance, a peptide partially purified from tamarind evidenced the reduction of food intake and weight gain in eutrophic Wistar rats, with the satiating effect being associated with increased CCK serum levels [83]. This peptide promotes the reduction of food consumption, as well as serum tumor necrosis factor alpha (TNF-α) and leptin levels, regardless of weight loss when administered to animals with obesity and metabolic syndrome induced by diet [84], together with a decrease in very-low-density lipoprotein (VLDL) and triglycerides (TG) [85]. Moreover, a potato extract (Potain) consisting of concentrated potato juice (a by-product of potato starch processing) containing 20% protein with trypsin-inhibitory activity suppressed food intake through CCK secretion. The authors suggest a direct stimulation of enteroendocrine cells and by luminal trypsin inhibition, although the active components responsible for this stimulation and the mechanism whereby they act have not yet been elucidated [86,87].

### 4.2. Carbohydrates

As outlined in previous sections, the IM plays an important role in the digestion and energy storage of host’s nutrients. Carbohydrates are a group of nutrients where bacterial digestion is key to provide metabolites with relevant activities in satiety signaling. An example of these compounds is SCFA (such as acetate, propionate, butyrate, and lactate), produced mainly by bacterial fermentation of dietary fiber and acting as ligands of G-protein-coupled receptors GPR41 and GPR43. These receptors, once activated, induce the expression of peptide hormones (such as leptin and PYY) that regulate appetite and energy metabolism and may promote nutrient absorption and the development of adipose tissue [40,88]. Some studies also suggest that IM could produce analogues of these neuropeptides, and thus it could have a much more direct role in regulating food intake [43]. There is also evidence that certain metabolic disorders associated with dysbiosis can be reversed. For instance, the transplantation of microbiota from lean or obese mice demonstrated that it could transfer their characteristics to a germ-free host [89]. Moreover, in *Helicobacter pylori*-infected mice that showed altered feeding patterns, delayed gastric emptying, and enhanced perception of satiety (coupled to increased release of postprandial cholecystokinin and increased expression of TNF-α), the eradication of *H. pylori* did not reverse the symptoms; however, subsequent administration of a *Lactobacillus* strain normalized the gastric function and feeding patterns [90].

Accumulated scientific evidence suggests that dietary interventions with molecules, such as certain prebiotic carbohydrates, capable of modulating the IM may be a valuable tool in the control of obesity and associated metabolic disorders [40,43]. Although it is possible to find multiple definitions for prebiotics, probably the most accepted is that of Gibson and Roberfroid, “nondigestible food ingredient that beneficially affects the host by selectively stimulating the growth and/or activity of one or a limited number of bacteria in the colon, and thus improves host health” [91]. While this definition includes different chemical compounds, the largest group is composed of plant-derived oligosaccharides. Fructooligosaccharides (FOS), galactooligosaccharides (GOS), arabinoxylan oligosaccharides (AXOS), or inulin-type fructans (ITFs) are amongst the best known and characterized. There is increasing evidence that these compounds, and hence oligosaccharide-rich diets, can exert a modulating effect on appetite and energy metabolism. For example, ITFs promoted the production of GLP-1 [40], which is involved in the reduction of appetite, fat mass, and hepatic insulin resistance [92]. Moreover, this prebiotic counteracts GPR43 overexpression in the adipose tissue, which is associated with a reduced number and size of adipocytes [93].

Furthermore, the administration of chicory root FOS improved metabolic disorders such as dyslipidemia, impaired intestinal permeability, metabolic endotoxemia, and diabetes in mice given high-fat diets through the regulation of the production and activity of peptides such as GLP-1 [42], and other gastrointestinal peptides such as PYY and ghrelin [86]. Indeed, the administration of this type of oligosaccharide reduced food intake and the development of fat mass and hepatic steatosis, both in normal rats and mice and in obese animals [94,95]. Treatments with wheat bran AXOS increased the level of circulating satietogenic peptides (PYY and GLP-1) and counteracted body weight gain and fat storage, in addition to reducing insulin resistance and improving the intestinal barrier [42]. Moreover, feeding GOS combined with *Lactobacillus rhamnosus* NCC4007 to mice colonized with microbiota from human infants reduced lipogenesis, the incorporation of triglycerides in lipoproteins, and triglyceride concentration in the liver and kidney [88]. Similar effects have been observed in human trials with healthy individuals. For example, in one trial with subjects fed a high-carbohydrate, low-fat diet, inulin administration led to reduced plasma triacylglycerol concentrations and reduced liver lipogenesis [96]. In other trial, treatment with FOS increased satiety and reduced food consumption [97], having a positive effect on reducing body mass index and total fat mass [98].

Oligofructose supplementation in overweight and obese adults showed the promotion of weight loss and improvement in glucose regulation in a randomized, double-blind, placebo-controlled trial. Suppressed ghrelin and enhanced PYY secretion were also observed, suggesting oligofructose may contribute to a reduction in energy intake compared to the placebo (maltodextrin) [99]. A treatment with inulin/oligofructose (50:50) in 30 obese women produced an increase in *Bifidobacterium* and *Faecalibacterium prausnitzii*, reducing serum LPS levels, while the prebiotics decreased *Bacteroides intestinalis*, *Bacteroides vulgatus*, and *Propionibacterium*, together with a slight decrease in fat mass [100]. The ability of prebiotics to modify the microbiota in children with overweight/obesity has been investigated, with results revealing that consumption of oligofructose-enriched inulin decreased body weight and body fat, significantly increasing *Bifidobacterium* spp. in fecal samples compared with children given placebo [101].

Other carbohydrates not considered as prebiotics have also shown the ability to influence the production of neuropeptides related to the gut–brain axis, modulating the feeling of satiety and food intake. In rats fed slow-digesting starch (SDS), expression of hypothalamic neuropeptide Y (NPY) and AgRP showed a significant reduction, and the anorexigenic corticotropin-releasing hormone (CRH) was increased, with a reduction in daily food intake [102]. Although further research is needed, these findings suggest that certain carbohydrates, especially some oligosaccharides, may have potential for the development of functional foods as a strategy for obesity control through modulation of the microbiota–gut–brain axis.

### 4.3. Lipids

The primary free fatty acid (FFA) sources are LCFA (long-chain FA) and MCFA (medium-chain FA), mainly derived from the hydrolysis of triglycerides by lipase digestion and SCFA produced by IM fermentation of indigestible dietary fiber. The unsaturated LCFA show multiple benefits for humans, with two main types: monounsaturated (MUFA) and polyunsaturated (PUFA). Omega-9 MUFA are non-essential, with oleic acid being one of the most representative. There are two types of PUFA, omega-3 and omega-6, which are essential in the diet and a substrate for intestinal microbiota [5]. The primary omega-3 fatty acids are α-linolenic acid (ALA), eicosapentaenoic acid (EPA), and docosahexaenoic acids (DHA), while linolenic acid (LA) is an example of omega-6 that metabolizes to arachidonic acid (AA). The recommended ratio of omega-6/omega-3 intake in the diet varies from 1:1 to 4:1, depending on many factors, including genetics and the health condition of the individual [103]. However, changes in nutritional habits that have occurred in recent years in the Western world have led to higher omega-6/omega-3 ratios, which have shown pro-thrombotic and pro-inflammatory effects, increasing the risk of developing obesity, diabetes, and other cardiovascular diseases. Omega-3 fatty acids play an essential role both in anti-inflammatory effects and in the prevention and control of obesity [104]. In terms of inducing satiety hormones secretion, studies indicate that a diet rich in PUFA increases CCK plasma levels in humans compared to a diet rich in MUFA, while both diets increase PYY levels [105,106]. Ligands for GRP40 and GPR120 include ALA, EPA, and DHA. Omega-3 FA, such as AA, activates GRP120 to the same extent as EPA and DHA. The difference between them is the kinetics and intensity of the activation, which are responsible for the different degree of CCK and GLP-1 secretion [104,107]. On the other hand, omega-3 FA strongly activates GPR120 and induces GLP-1 and CCK release in STC-1. ALA, found in green leafy vegetable chloroplasts, flaxseed, hemp, walnut, soybean, and canola oil, could activate GRP40 and induce CCK release. DHA is the most potent GPR40 agonist among the unsaturated FA, mediating in the upregulation of GIP, GLP-1, and CCK. Both DHA and EPA are in fish and fish oil, particularly in bluefish, as well as in shellfish [30]. Intestinal bacteria can ferment LA and ALA into other FAs that have shown to stimulate intestinal hormones secretion [29].

Nuts such as pistachios, walnuts, macadamia nuts, almonds, hazelnuts, pecans, cashew nuts, Brazil nuts, Korean pine nuts, and chestnuts, containing significant amounts of MUFA and PUFA, have shown a weight loss and satiating effect in numerous studies [108,109]. Nonetheless, there is experimental evidence for some of these foods that demonstrates increased stimulation of intestinal hormone release through EECs. Peanuts, with a high MUFA and PUFA content, have also demonstrated stimulation of satiety feelings. The effect of pistachios on weight control and food intake is related to increased GLP-1 and GIP levels [108]. Pieces of evidence report an increase of GLP-1 secretion by walnut oil, whereas there is controversy as to whether it also influences CCK and PYY levels [110]. A randomized cross-over clinical trial with obese women proved the stimulating effect on intestinal hormone release of peanuts, showing an increase of GIP, GLP-1, and CCK levels [111]. Reports on Korean pine nut oil indicate a significant increase of CCK-8 and GLP-1 in humans [109]. Other omega-3 rich foods such as oat oil have shown increased and prolonged PYY, GLP-1, and CCK release in humans. In contrast, although MFCAs suppress food intake [111], their ability to induce intestinal peptide hormone secretion still needs to be further confirmed. 

Finally, acetate, propionate, and butyrate, the most abundant SCFAs, have many essential functions in the energetic metabolism and the intestinal immune homeostasis, as well as having protective effects from diet-induced obesity. Particularly propionate and butyrate induce intestinal hormone secretion, decreasing food intake and increasing satiety feelings. On the contrary, increased levels of acetate derivate from dysbiosis activated parasympathetic nervous system increased insulin levels induced by glucose and increased ghrelin levels, which caused hyperphagia and obesity [16,29]. Furthermore, FFAR2 exhibits a higher preference for shorter chains, C2=C3>C4> other SCFA, according to their potency. Even though propionate and butyrate activate FFRA3, the receptor shows a higher response to larger chains [29]. Recently, these receptors were described in vagal afferent neurons, where SCFA can directly activate the neural pathway [112]. Dietary sources of SCFA are not only nondigestible carbohydrates, but also sourdough bread, vinegar and vinegar-based products such as pickles, and some cheeses and other dairy products [113]. Both rich dietary sources of SCFA and physical exercise might play a key role on the prevention and treatment of many disease such as obesity and metabolic syndrome [114].

## 5. Challenges of In Vitro Models to Study the Microbiota–Gut–Brain Axis Regulation

The complexity of the mechanisms involved in the microbiota–gut–brain axis have been discussed throughout this review. The study of this complex scenario requires reliable in vitro tools to recapitulate the human physiology, allowing the improvement of the scientific knowledge and also the development of new therapeutics. 

2D cell models consisting of a single layer of cells (monolayer) growing on a surface (plastic, membrane) represent a simplified tool to study interactions in a controlled manner, providing access to both sides: apical and basolateral. Most of them involve only one type of cell or tissue cultured in isolation. For instance, Caco-2 cell line, derived from human colorectal adenocarcinoma, has been used to study the intestinal absorption of drugs. They are usually cultured on semi-permeable membranes in a transwell format, where they spontaneously differentiate, after 21 days in culture, into cells possessing the morphology and function of enterocytes, the absorptive cells of the intestine [115]. The STC-1 cell line, derived from mouse small intestinal neuroendocrine carcinoma, has been used in numeral screening studies (in vitro) to identify bioactive components in food involved in the regulation of hormones secretion. STC-1 cells present features of native intestinal enteroendocrine cells and secrete a wide range of gut hormones, such as CCK, GIP, PYY, GLP-1, and OXM [116]. Due to limitations associated with cell lines, many studies have focused on the use of primary cells, as a more physiologically relevant cell-based model. A 2D co-culture system of primary intestinal epithelial cells and primary enteric neurons and glia was developed to study the neurointestinal crosstalk. The results demonstrate the importance of enteric populations in the regulation of intestinal barrier function [117]. Co-cultured (2D) monolayers of human intestinal stem cell-derived enteroid and human monocyte-derived macrophages were tested to explore epithelial and macrophage interactions through the evaluation of barrier function, cytokine secretion, and protein expression under basal conditions and following bacterial infection. A coordinated epithelial–immune cell response to pathogens was demonstrated [118]. The use of primary cells is often restricted by limited donors and low viability in culture conditions. Despite the valuable contributions of 2D models to the biomedical research, they often failed the prediction of in vivo responses [119]. The understanding of human physiology and deregulation of cellular behaviors is crucial for disease prevention, diagnostics, and therapeutics. 

Higher levels of cell differentiation and tissue organization have been achieved by 3D tissue models, using encapsulation of cells within hydrogels [120] or organoid technology that is based on 3D clusters of primary or stem cells [121]. Despite their capacity to closer emulate the in vivo physiology, the lack of vasculature and the static conditions hampers the continuous transport of soluble factors and the removal of metabolic products from the tissue. 

Organ-on-a-chip (OoC) technology represents a suitable approach to model multi-organ communication on the basis of three-dimensional cell cultures and microfluidic channels and chambers able to recreate the microphysiological environment. Multi-chambers are connected, allowing the diffusion of soluble factors and/or metabolites to mimic cross-communication in living organs. The fluid’s flow inside the microfluidic channels creates shear stress suitable to stimulate cell proliferation and differentiation. OoC offers the possibility to apply mechanical forces to mimic the physical microenvironment of living organs, such as peristaltic movements, and also the integration of sensors to monitor and even control of biological and physical parameters, such as pH, transepithelial electrical resistance (TEER), oxygen pressure, metabolite production, and signal transmission [121].

Moreover, the scale-down reduces the number of reagents needed. The technology empowers drug development, safety assessment, disease modelling, and personalized medicine. Moreover, the modular approach allows for the coupling of devices in a sequential mode to evaluate the crosstalk between different organs [122,123]. Although different organs-on-a-chip are already in development in several laboratories around the world, the path to model the whole MGBA through a multiorgan-on-a-chip approach is still long overdue for the in vivo complex mechanisms [124]. The recent European Research Council-funded project named MINERVA, aimed at developing an innovative technological platform to investigate the relationship between the intestinal microflora and brain functionality in healthy and pathological conditions, might represent a significant contribution to the field.

## 6. Conclusions

Complex mechanisms regulate energy homeostasis and satiety, where intestinal hormones have a central physiological function as signaling systems. Food induces satiety in the intestine through mechanical stimulation and by sensing of the luminal contents by enteroendocrine cells (EECs), which also involves the release of hormones. Scientific evidence also supports the fact that intestinal microbiota interplays in the regulation of food intake and satiety. This modulatory activity is due to the interaction of microbiota with EECs, or through its bioactive metabolites on the intestinal lumen by releasing chemosensing factors. The diet is known to modulate the structure of the gastrointestinal microbiota by providing specific nutrient sources and inducing environmental changes in the gut ecosystem. There is, therefore, an interplay between diet and microbiota–gut–brain axis regulation with a pivotal role in the control of satiety. Owing to current non-invasive treatments lacking efficacy in combating obesity, food-derived bioactive compounds are proposed as endocrine-modulatory molecules stimulating intestinal hormone release to promote satiety feelings and suppress food intake. Dietary interventions based on the intake of certain nutrients or their metabolites is an alternative for developing nutraceuticals or functional foods targeting obesity and other metabolic diseases. For this purpose, we require further research to understand the mechanisms underlying the endocrine modulatory activity of nutrient-derived bioactives and how they can positively modulate the microbiota–gut–brain axis. Organ-on-a-chip technology represents a promising platform for the screening of these compounds through it being a suitable approach to model multi-organ communication such as the microbiota–gut–brain axis.

## Figures and Tables

**Figure 1 nutrients-13-00632-f001:**
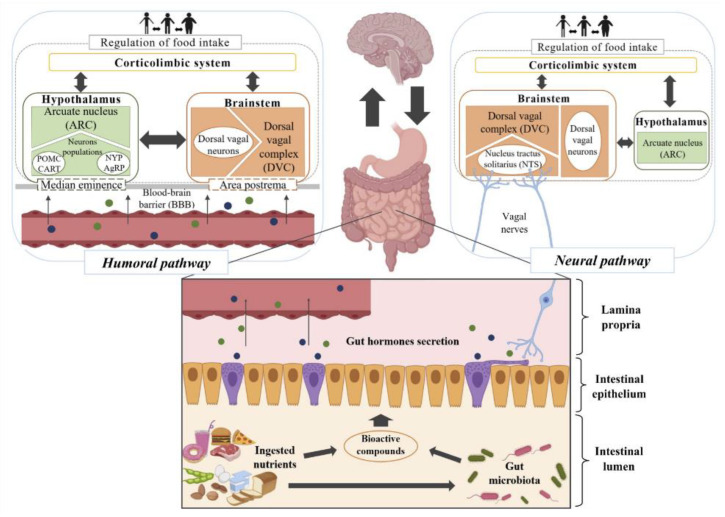
Scheme of the humoral and neural pathways connecting gut hormone secretion by enteroendocrine cells (EECs) and the different brain regions.

**Figure 2 nutrients-13-00632-f002:**
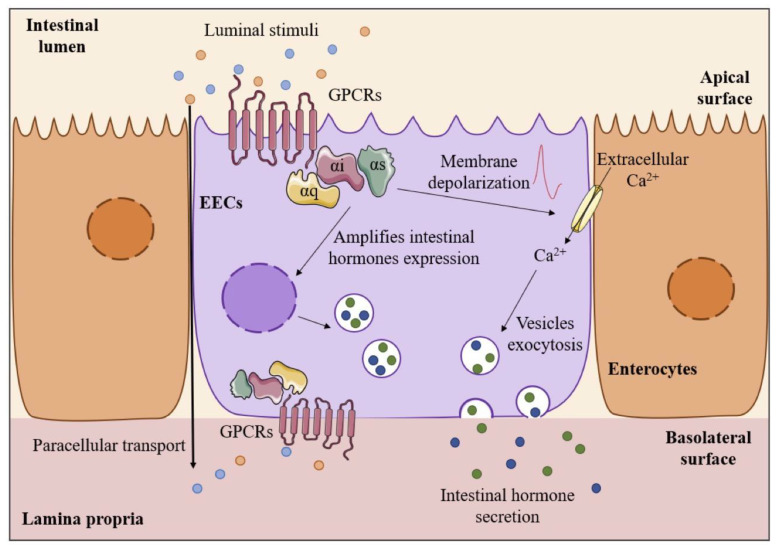
Scheme of EECs as sensors of the luminal intestinal content through G-protein-coupled receptors (GPCRs).

**Figure 3 nutrients-13-00632-f003:**
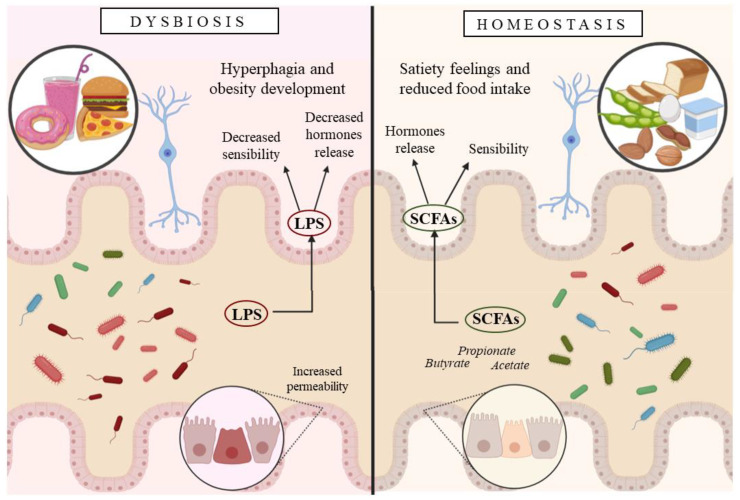
Scheme illustrating the mechanism through which the gut microbiota affects the host.

**Table 1 nutrients-13-00632-t001:** Summary of the main G-protein-coupled receptors (GPCRs).

Receptors	Cell Expression	Intestinal Location	Luminal Ligands	Coupled G-Proteins	Outcome	Reference
GPRC6A	L cells	Widely expressed in GIT	Basic L-amino acids (L-Lys, L-Arg, and L-Orn), sulfur-containing L-amino acids, and divalent cations	Gα_s_, Gα_i_, and Gα_q_	GLP-1 exocytosis	[10,19]
T1R1/ T1R3	I cells	Duodenum	Natural sugar, artificial sweeteners, L-amino acids, and monosodium glutamate (potential by IMP and GMP)	Gα_i_ (α- gustducin and α-transducin)	GLP-1 and CCK exocytosis	[20,21]
GPR142	K and L cells ^1^	Small intestine	Aromatic amino acids such as L-Phe and L-Trp	Gα_q_	GLP-1 and GIP exocytosis	[22,23]
GRP93 (GPR92/ LPARS)	I and L cells	Duodenum	Peptones (luminal protein hydrolysate)	Gα_q_ and Gα_i_	Amplifies CCK expression and CCK exocytosis	[24,25]
PepT1	L cells	Small intestine to the colon	Oligopeptides, and di- and tripeptides	-	GLP-1 and CCK ^2^ exocytosis	[17]
CaSR	L and K cells	Caecum	Peptides, oligopeptides, and L-amino acids, preferably aromatic and aliphatic; ion calcium and polyamines	Gα_q_ and Gα_i_	CCK and, in vitro, also GLP-1, GIP, and PYY exocytosis	[17,21,26]
FFAR1/ FFAR4	I, K, and L cells	Ileum and large intestine	MCFA and LCFA	Gα_i_ and Gα_s_/Gα_s_, Gα_i_, and Gα_q_	GLP-1, CCK, and GIP exocytosis	[24,27,28,29]
FFAR2/ FFAR3	L cells	Ileum and large intestine	SCFA	Gα_i_ and Gα_q_/Gα_i_	GPL-1 and PYY exocytosis	[5,28,30,31]
GPR119	K and L cells	Ileum	Oleoyl ethanolamide	Gα_s_	GLP-1 and GIP exocytosis	[10,22,32]

^1^ Suggested by preliminary studies [23]; ^2^ previous studies support that the activation of PepT1 leads to CCK_1_ receptor-sensitive vagal afferent activation by a paracrine stimulus that indirectly mediates CCK secretion in I cells [12,33].

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
