# Peer review of "A Review on the Role of Food-Derived Bioactive Molecules and the Microbiota–Gut–Brain Axis in Satiety Regulation"

_nutrients, 2021, doi:10.3390/nu13020632_

Round 1

Reviewer 1 Report

The paper provides good overview of the nutrients effects on intestinal satiety signaling. Weaker part is the role of microbiota, in fact the discussion is limited to SCFA, however there are solid data implicating bacterial proteins in signaling satiety, ex PMID: 31911661, PMID: 31491982 , linking nutrients with gut microbiota. Please discuss.

Author Response

Thank you for identifying this aspect on the effect of the bacterial proteins in the gut-brain axis regulation that improved the review content. The two papers are now cited in the text and their results discussed (please see lines 184-193).

Reviewer 2 Report

Line 35

Clarify what you mean by “imbalanced diet”

Line 47-50

Please unify the two sentences to describe the interest of food-derived compounds

Line 70

Clarify what you mean by “an order (efferent pathway)”

Line 138-179

This section should be more extensive. Add a figure of the proposed mechanisms through which the gut microbiota affects the host or the potential regulation of hunger perception and satiety (dysbiosis-SCFA production).

Line 140

Add a comma after “vitamins”

Line143

Change “contribute to the development of different diseases” by “could contribute to the pathogenesis and/or progression of diseases”

Line 168-170

“bacterial loss” or maybe “microbial diversity”

Mention also antibiotics as dysbiosis treatment

Line 177

Add more references

Line 196-197

Change “there is a bidirectional…” by  “there could be a bidirectional…”

Line 201-202

Please, add a reference about this evidence

Line 243

Describe the sample studied “in treated subjects”

Line 261

Delete a space between “find” and “correlation”

Line 266-268

Rewrite this sentence, it is confusing

Line 273-274

Please describe through which mechanism

Line 293

Mention also the role of SCFA on colonic cell health

Line 275-343

Please, revise studies about the role of food-derived bioactive compounds (such as Inulin, FOS…) on intestinal hormone release in humans according to different pathologies (obese, non-obese, type 2 diabetes,…) and add results in this section

Include a recent systematic review about the effects of probiotics on appetite-related hormones (PMID:33143930)

Line 353

Check the ratio of omega6/omega 3 and add reference

Line 362-363

Try to complete information or delete sentence

Line 384

Delete a space after “obesity,”

Line 385-388

This statement is misleading, please rewrite

Line 391-393

Please describe if dietary sources of SCFA are relevant respect to endogen production

Check reference 103 (in short or maybe medium-chain triglycerides effects?)

Author Response

Question 1:

Line 35

Clarify what you mean by “imbalanced diet”

 Answer 1:

We understand an imbalanced diet is the ingestion of foods containing more or fewer nutrients, either proteins, carbohydrates, lipids or vitamins than those of a healthy/balanced diet according to the WHO, https://www.who.int/news-room/fact-sheets/detail/healthy-diet). For instance, an imbalanced diet can be the increased intake of foods with a high content in fat and sugars that contributes to a higher energy intake, which together with a reduced caloric expenditure, can lead to fat accumulation and, if sustained, to overweight and obesity.

In summary, what we mean in this sentence is that the sustained consumption of an imbalanced diet can lead to overweight and obesity but we can rephrase the sentence or replace “imbalanced diet” by “unhealthy diet”, if the reviewer finds it more appropriate.

Question 2:

Line 47-50

Please unify the two sentences to describe the interest of food-derived compounds

 Answer 2:

We have unified these sentences and now the text is as follows: “These appetite inhibitors or satiety stimulators have attracted considerable interest in addressing new treatments for obesity and overweight.            Particularly, food-derived compounds are regarded in the literature as interesting molecules to take advantage of their own natural and innate effects on food intake regulation.” (please see lines 46-49).

Question 3:

Line 70

Clarify what you mean by “an order (efferent pathway)”

 Answer 3:

The central nervous system transmits impulses to the periphery causing an effect or action, so those signals are received as an order. We propose to clarify the sentence as follows: “The intestinal hormones, including cholecystokinin (CCK), glucagon-like peptide 1 (GLP-1), peptide YY (PYY), oxyntomodulin (OXM), and gluten immunogenic peptides (GIP) arrive at the brain (afferent pathway) and generate central signals (afferent pathway) that transmit impulses to the peripheral organs causing an effect (efferent pathway).” Please see lines 66-69 for clarification.

Question 4:

Line 138-179

This section should be more extensive. Add a figure of the proposed mechanisms through which the gut microbiota affects the host or the potential regulation of hunger perception and satiety (dysbiosis-SCFA production)

 Answer 4:

We have enlarged the section about the role of the microbiota in the regulation of the gut-brain axis, including some relevant papers that give a more critical view of the contribution of the microbiome towards satiety, obesity and other related cardiovascular diseases.

According to your suggestion, we have also produced a figure illustrating the proposed mechanisms for regulation of hunger and satiety, and the role of microbiota and their metabolites (SCFA).

Question 5:

Line 140

Add a comma after “vitamins”

Answer 5:

Thank you for spotting the mistake, we have added a comma.

Question 6:

Line143

Change “contribute to the development of different diseases” by “could contribute to the pathogenesis and/or progression of diseases”

Answer 6:

We have changed the sentence according to your suggestion.

 Question 7:

Line 168-170

“bacterial loss” or maybe “microbial diversity”

Mention also antibiotics as dysbiosis treatment

Answer 7:

We changed bacterial loss to microbial diversity, according to your suggestion.

Also a reference to the contribution of antibiotics to dysbiosis was included (please see line 42).

 Question 8:

Line 177

Add more references

Answer 8:

Two more recent references about the practice of using probiotics and prebiotics in different gastrointestinal diseases and health conditions have been added (please see line 205, and references 55 and 56).

 Question 9:

Line 196-197

Change “there is a bidirectional…” by  “there could be a bidirectional…”

Answer 9:

We have corrected the sentence according to your suggestion.

 Question 10:

Line 201-202

Please, add a reference about this evidence

  Answer 10:

We have improved the sentence and include two new references (see references 62 and 63).

Question 11:

Line 243

Describe the sample studied “in treated subjects”

 Answer 11:

We have given details about the supplemented diet utilized in this study (see lines 277-278).

 Question 12:

Line 261

Delete a space between “find” and “correlation”

 Answer 12:

We have removed the space.

 Question 13:

Line 266-268

Rewrite this sentence, it is confusing

 Answer 13:

The sentence was improved by reorganising and omitting the irrelevant information (see lines 302-304).

 Question 14:

Line 273-274

Please describe through which mechanism

 Answer 14:

The studies that report the food intake suppression mediated by CCK secretion of the concentrated potato juice extract Potain, concluded there is a direct stimulation of enteroendocrine cells since in vitro digestion by pepsin and pancreatin did not reduce the CCK-releasing activity of Potein. However the author did not identify the active components responsible for the stimulation, neither proposed a mechanism.

To complete this piece of information, now the text is as follows: “Also, a potato extract (Potain) consisting of concentrated potato juice (a by-product of potato starch processing) containing 20% protein with trypsin-inhibitory activity suppressed food intake through CCK secretion. The authors suggest a direct stimulation of enteroendocrine cells and by luminal trypsin inhibition, although the active components responsible for this stimulation and the mechanism whereby they act have not yet been elucidated [86,87].” Please see lines 309-311.

 Question 15:

Line 293

Mention also the role of SCFA on colonic cell health

 Answer 15:

The anti-inflammatory role of SCFA in the colon is already mentioned in lines 175-183.

 Question 16:

Line 275-343

Please, revise studies about the role of food-derived bioactive compounds (such as Inulin, FOS…) on intestinal hormone release in humans according to different pathologies (obese, non-obese, type 2 diabetes,…) and add results in this section

Include a recent systematic review about the effects of probiotics on appetite-related hormones (PMID:33143930)

Answer 16:

We have discussed results related to supplementation of diets with prebiotics in satiety response indifferent groups of people with obesity/ overweight. Through our literature search it was difficult to identify specific results related to the direct effect of the prebiotic compounds on the intestinal hormone secretion. For this reason, only minor references to this particular topic are outlined in the text. Please see lines 359-269.

Reference to the systematic review suggested by the reviewer can be found in lines 205-211.

Question 17:

Line 353

Check the ratio of omega6/omega 3 and add reference

 Answer 17:

We have checked the ratio omega 6/omega 3 and corrected the text accordingly. Now the text is as follows: “The recommended ratio of omega-6/omega-3 intake in the diet varies from 1:1 to 4:1 depending on many factors, including genetics and the health condition of the individual [103]. However, changes in nutritional habits that have occurred in recent years in the Western world have led to higher omega 6/omega 3 ratios, which have shown pro-thrombotic and pro-inflammatory effects, increasing the risk of developing obesity, diabetes and other cardiovascular diseases.” Please see lines 388-392.

 Question 18:

Line 362-363

Try to complete information or delete sentence

 Answer 18:

The sentence was completed according to the reviewer suggestion (see line 401).

 Question 19:

Line 384

Delete a space after “obesity,”

 Answer 19:

We have removed the space.

 Question 20:

Line 385-388

This statement is misleading, please rewrite

Answer 20:

The sentence was improved (please see lines 429-434).

 Question 21:

Line 391-393

Please describe if dietary sources of SCFA are relevant respect to endogen production

Answer 21:

 Rich-dietary sources of SCFA have been shown to exert multiple beneficial effects on mammalian energy metabolism that it is largely caused by the endogen effect. Focused on satiety and obesity, the increased of anorexigenic hormone releases leading by the increased not only of rich-dietary sources of SCFA but also of physical exercise.

Question 22:

Check reference 103 (in short or maybe medium-chain triglycerides effects?)

Answer 22:

Thank you for spotting the mistake. The reference applied to medium-chain triglycerides and was removed.

Submission Date

01 December 2020

Date of this review

21 Jan 2021 13:51:05

Round 2

Reviewer 2 Report

The mansucript has been significantly improved. Thanks